# Gene Profiling in the Adipose Fin of Salmonid Fishes Supports Its Function as a Flow Sensor

**DOI:** 10.3390/genes11010021

**Published:** 2019-12-23

**Authors:** Raphael Koll, Joan Martorell Ribera, Ronald M. Brunner, Alexander Rebl, Tom Goldammer

**Affiliations:** 1Fish Genetics Unit, Institute for Genome Biology, Leibniz Institute for Farm Animal Biology (FBN), Wilhelm-Stahl-Allee 2, 18196 Dummerstorf, Germany; raphaelkoll@gmx.de (R.K.); martorell-ribera@fbn-dummerstorf.de (J.M.R.); brunner@fbn-dummerstorf.de (R.M.B.); rebl@fbn-dummerstorf.de (A.R.); 2Professorship for Molecular Biology and Fish Genetics, Faculty of Agriculture and Environmental Sciences, University of Rostock, 18055 Rostock, Germany

**Keywords:** adipose fin, fin-clipping, welfare, *Oncorhynchus mykiss*, *Coregonus maraena*, salmonids, mechanoreceptors, innervation

## Abstract

In stock enhancement and sea-ranching procedures, the adipose fin of hundreds of millions of salmonids is removed for marking purposes annually. However, recent studies proved the significance of the adipose fin as a flow sensor and attraction feature. In the present study, we profiled the specific expression of 20 neuron- and glial cell-marker genes in the adipose fin and seven other tissues (including dorsal and pectoral fin, brain, skin, muscle, head kidney, and liver) of the salmonid species rainbow trout *Oncorhynchus mykiss* and maraena whitefish *Coregonus maraena.* Moreover, we measured the transcript abundance of genes coding for 15 mechanoreceptive channel proteins from a variety of mechanoreceptors known in vertebrates. The overall expression patterns indicate the presence of the entire repertoire of neurons, glial cells and receptor proteins on the RNA level. This quantification suggests that the adipose fin contains considerable amounts of small nerve fibers with unmyelinated or slightly myelinated axons and most likely mechanoreceptive potential. The findings are consistent for both rainbow trout and maraena whitefish and support a previous hypothesis about the innervation and potential flow sensory function of the adipose fin. Moreover, our data suggest that the resection of the adipose fin has a stronger impact on the welfare of salmonid fish than previously assumed.

## 1. Introduction

Salmonid fishes, including rainbow trout *Oncorhynchus mykiss* (Walbaum, 1792), Atlantic salmon *Salmo salar* L., and maraena whitefish *Coregonus maraena* (Bloch, 1779) are farmed in aquaculture facilities all over the world [1]. Their common characteristic is the adipose fin, which is situated on the dorsal midline between dorsal and caudal fin, although a total of 6000 species from eight orders of the Teleostei all possess an adipose fin [2]. Large numbers of artificially bred juvenile salmonids are released into sea-ranching procedures every year to produce 4.4 million tons of top-class food fish [3]. Furthermore, billions of salmonids are released in restocking or stock-enhancement projects [4]. Most of these animals are tagged to monitor the success of those research projects or indicate ownership relations [5] and to identify escapees from aquaculture farms. Those are considered as a serious problem since they reduce the natural gene pool [6]. 

In order to determine the most appropriate method suitable for routine large-scale screenings of all salmonids bred in Norwegian aquaculture systems, the Panel on Animal Health and Welfare of the Norwegian Scientific Committee for Food Safety evaluated all available marking techniques in 2016. These comprised (i) externally attached visible tags, (ii) visible internal tags, (iii) chemical marking, (iv) remotely detectable internal tags, (v) freeze branding, and (vi) fin clipping. The clipping of fins, especially of the adipose fin, was found to be the most applied and was evaluated as the only persistent and cost-efficient technique available. Unlike other fin structures [7], the adipose fin does not regrow when clipped completely [8,9,10,11]. In addition, fin clipping compromises the welfare of the fish [12]. Nonetheless, in European countries, such as Sweden, Estonia, and Latvia, all hatchery-reared salmon are mandatorily marked by adipose-fin clipping to facilitate the differentiation of farmed fish from natural stocks [13]. Adipose-fin clipping of Pacific salmon species is performed on a much larger scale. In the State of Washington (US) alone, more than 200 million juvenile salmonids are adipose fin-clipped every year [14]. Dozens of recapture studies reveal inconclusive influences on the growth and survival of fin-clipped animals [8,9,10,11,15,16,17,18,19,20,21]. Noteworthy in the context of fish welfare is that the resection of the adipose fin significantly reduces the swimming efficiency of *O. mykiss* juveniles in a flowing current [22]. Subsequent studies proved the innervation of the adipose fin in brown trout *Salmo trutta* [23] and a mechanoreceptive function of the adipose fin in catfish *Corydoras aeneus* [24]. These studies underscore that the adipose fin is not a useless body appendage, as originally claimed [25], but a mechanosensor contributing to optimal swimming performance [26]. 

Fin-clipping not only removes a supposedly useful organ. It can be assumed that the process itself causes pain. Nowadays, it is indisputable that fish are sentient beings [27,28,29,30], at the latest since damage- and pain-signaling nociceptors have been discovered in *O. mykiss* [27,29,30,31].

Somatosensory perception involves the activation of primary sensory neurons, whose somas reside within the dorsal root ganglia (DRG) or cranial sensory ganglia in the head region of the lateral line system [32,33,34] (Figure 1). The DRG neurons are pseudo-unipolar [33]. The axon has two branches, one penetrating the spinal cord to synapse with central nerve-system (CNS) neurons, and the other forms free peripheral endings or associates with peripheral targets. They respond to a wide range of stimuli comprising noxious mechanical or thermal stimuli as well as different kinds of touch [33,35]. Previous studies on higher vertebrates based on single-cell RNA-seq [35,36,37,38,39,40,41,42,43,44,45,46,47] and immunohistology [48,49,50,51,52,53,54] have identified particular sets of genes that indicate either specific sections and/or specific functions of the neuron and glial cells. The discovery of local mRNA translation within the axon outside the neuronal soma (reviewed in [55]) allows further analysis of the quality and functions of the nerves. All relevant genes selected in this study were shown to be present within the axon of sensory neurons (supplementary materials of [41]). 

In order to evaluate the influence of the adipose-fin resection based on measurable and thus objective criteria, we profiled the expression of a panel of 35 genes in the adipose fins (AF) of *O. mykiss* and *C. maraena*. The obtained qPCR data were compared against the expression in a range of further tissues, including dorsal and pectoral fin, brain, skin, muscle, head kidney (HK), and liver. 

## 2. Materials and Methods 

### 2.1. Fish and Sampling

Juvenile *O. mykiss* of the German selection strain BORN at the age of 15 months (*n* = 7, 26.24 ± 0.78 cm, 302.71 ± 42.78 g) were selected for this analysis, as these salmonids are naturally adapted to flow regime. Fish were kept in a flow-through aquaculture system of the State Research Centre for Agriculture and Fisheries (LFA-MV). Additionally, we chose *C. maraena* (*n* = 3, 17.5 ± 1.08 cm, 64.37 ± 10.88 g) as a second salmonid species for our investigations, also kept in recirculating aquaculture systems from the LFA-MV. After stunning and killing fish by electrical flow, brain, muscle, skin, HK, and liver were sampled. In addition, we resected AF, dorsal fin, (DF), and pectoral fin (PF) as entire target tissues without removing the skin. Samples were immediately transferred to liquid nitrogen and stored at −80 °C until further processing. 

The experimental protocol was approved by the Committee on the Ethics of Animal Experiments of Mecklenburg-Western Pomerania (Landesamt für Gesundheit und Soziales LAGuS; approval ID: 7221.3-1-012/19).

### 2.2. Gene Selection and Primer Design

We selected 35 genes, which have been described as selectively expressed by either nerve cells, glial cells, or receptor corpuscles. Unlike mammals, the common ancestor of the extant salmonid species underwent an additional teleost- and an additional salmonid-specific round of whole-genome duplication (WGD) [1,56]. These two events multiplied the number of particular genes in, for instance, *O. mykiss* and *C. maraena*. The WGD-derived paralogous genes are known as ohnologs [56], but their individual functions are largely unknown yet and it is to be expected that they are expressed to varying degrees. Our designed primer pairs detect either multiple paralogs/ohnologs or specifically a particular paralog/ohnolog. These details are listed together with the accession numbers and putative functions of the 35 target genes in Appendix A, Table A1. BLAST searches were performed using the NCBI server to identify possible gene duplicates and transcript variants in salmonids. All identified sequences were aligned using the Clustal Omega Multiple Alignment tool. Gene-specific oligonucleotides were designed applying Pyrosequencing Assay Design software v.1.0.6 (Biotage). The same primer pairs were used for the quantification of cDNA samples from *O. mykiss* and *C. maraena*.

### 2.3. RNA Preparation and cDNA Synthesis

The tissues were homogenized and RNA was isolated using Trizol (Life Technologies–Thermo Fisher Scientific, Karlsruhe, Germany), followed by a purification with the RNeasy Mini Kit (Qiagen, Hilden, Germany) with 15 min in-column DNase treatment. Spectrophotometry (NanoDrop One, Thermo Fisher Scientific, Karlsruhe, Germany) and gel electrophoresis were used to evaluate the quality and quantity of the isolated RNA. SuperScript II Reverse Transcriptase Kit (Thermo Fisher Scientific, Karlsruhe, Germany) was used to generate cDNA according to the manufacturer’s instructions. 

### 2.4. Gene-Expression Profiling via qPCR

Quantitative real-time PCR (qPCR) was carried out on the LightCycler 96 system (Roche, Basel, Switzerland) to detect and quantify specific transcript amounts. The LightCycler protocol used was optimized for a 12-µl reaction volume. A ready-to-use SensiFAST SYBR No-ROX Mix (Bioline, Luckenwalde, Germany) was mixed with the cDNA aliquots and applied to Light Cycler 480 Multiwell 96 plates (Roche). The qPCR program included an initial denaturation (95 °C, 5 min.), followed by 40 cycles of denaturation (95 °C, 5 min.), annealing (60 °C, 15 s) and elongation (72 °C, 15 s) steps and the fluorescence measurement (72 °C, 10 s). All melting curves were inspected to validate the absence of unspecific amplicons. In addition, PCR products were visualized on agarose gels to assess product size and quality. Individual copy numbers were calculated based on external gene-specific standard curves (10^7^–10^3^ copies per 5 μL). To control for variations in isolation, reverse-transcription yield, and amplification efficiency [57], the obtained copy numbers were then normalized with a factor based on the geometric mean of the three reference genes *EEF1A1*, *RPS5* and *18S* (*O. mykiss*) and *RPL9*, *RPL32,* and *EEF1A1b* (*C. maraena*), respectively [58,59,60]. 

Due to the lack of sequence information regarding transcript variants of *C. maraena*, the amplicons of the *C. maraena* genes were sequenced. Sequencing was performed with qPCR primers using the ABI BigDye Terminator v3.1 Cycle Sequencing Kit and ABIPrism DNA sequencer (Applied Biosystems, Waltham, MA, USA), following the modified Sanger method [61].

### 2.5. Data Analysis

All data were evaluated for statistical significance using IBM SPSS Statistics 25. Global analysis of variance or Kruskal–Wallis H-Test was used with subsequent post-hoc tests. In all tests, a *p*-value of ≤ 0.05 indicated significance. Standard error of the mean (SEM) was calculated as described by [62]. 

## 3. Results

Sequences from 37 genes (including orthologue variants) were identified for *O. mykiss* and *C. maraena,* and expression profiling was performed in eight selected (strongly and weakly innervated) tissues. The gene panel was divided into two sets. Set 1 contained genes that indicate the presence of nervous cells, particularly those that are expressed exclusively in the neuronal axon, dendrites, or nucleus. Set 2 contained genes that indicate the presence of specific receptor structures. In general, expression of all analyzed marker genes was detectable in the adipose fin, often exceeding the expression levels of other nerve-traversed tissues. The presentation in this section is limited to those genes that have been identified as informative markers in previous studies and that showed significant differences in expression between tissues in the present study. (Data on other genes are given in Figure 4a and Appendix A
Figure A1).

### 3.1. The Adipose Fin Showed High Levels of Neuron Marker Expression

The neuron-marker genes *NEFL*, *PRPH*, *PVALB7*, *NGFR*, *GFAP*, *MPB*, *MPZ* (Figure 2), and *PMP22*, *S100B*, *NCAM1*, *SOX10* (Figure A1) were detectable at high levels (between 1 × 10^3^ and 4 × 10^6^ transcripts per 1 µg RNA) in the three fin types (AF, DF, and PF) investigated in the rainbow trout and maraena whitefish (Figure A2). In most cases, the expression represented only a fraction of that detected in brain samples. On the other hand, the expression levels of all above-mentioned neuron marker genes (except for *PVALB7* in the fins) significantly surpassed the expression in the liver by >4.5-fold (*PRPH*) to >210.3-fold (*NCAM1*) and in the HK by >1.2-fold (*NCAM1*) to >37.2-fold (*MBP*). Particularly, the genes coding for neurofilament light polypeptide (NEFL) and neurofilament 4 (PRPH) (markers for small axons) showed significantly higher mRNA abundances in AF compared with skin (18.4 and 4.4-fold higher), muscle (6.4- and 1.3-fold higher), liver (41.2- and 2.9-fold higher) and HK (8.6- and 1.7-fold higher). Noteworthy, the gene encoding the high-affinity calcium ion-binding protein parvalbumin (PVALB7), a marker gene for large axons, was highly expressed (>1 × 10^5^ copies/µg RNA) in skin, muscle, and brain (Figure 2). 

*NGFR* transcripts encoding the neurotrophic receptors, which characterize types of neurons and neuron-associated glial cells, were detected at high levels (>1 × 10^5^ copies/µg RNA) in brain, skin, muscle, and the three fin types, while it was virtually absent in liver and HK (Figure 2). Moreover, the *NGFR* copy number was >6.2-fold higher in AF compared with the copy number in skin, liver, and HK and even exceeded the values in brain samples. The glial-cell marker genes *GFAP*, *MBP*, *MPZ* (Figure 2), *PMP22*, *S100B*, *NCAM1*, and *SOX10* (Figure A1a) were highly expressed (from >2 × 10^4^ to >5.7 × 10^6^ copies/µg RNA) in the brain, as expected. Particularly, the gene encoding the glial fibrillary acidic protein (*GFAP*) was strongly expressed in the brain but was also detectable in AF and DF in substantial levels. The genes coding for the myelin-forming MBP, MPZ (Figure 2), and PMP22 (Figure A1a) were strongly expressed in the skin and to a lesser but still remarkable extent in the fins. The general neuron and glial-cell marker genes *S100B*, *NCAM1*, and *SOX10* (Figure 4a, Figure A1a) were strongly expressed in brain, skin, muscle, and the three fins investigated, but merely detectable in the liver. 

### 3.2. Genes Coding for Mechanoreceptor Proteins were Expressed in the Adipose Fins

The receptor marker genes *TRPC1*, *ASIC1*, -2 and -4, *KCNK2* and -4 and *PIEZO2* showed distinct expression patterns in the investigated tissues (Figure 3 and Figure 4b). TRPC1 is a mechanoreceptive channel protein, whose mRNA level was extremely high in brain (~850,000 copies/µg RNA), followed by AF (~10,000 copies/µg RNA) and muscle (~8000 copies/µg RNA) (Figure 3). The genes coding for the mechanoreceptive potassium channel protein KCNK2, -4 and -10 were most strongly expressed in the brain (>2500 copies/µg RNA). Among the *KCNK* genes, *KCNK2* revealed the highest transcript abundance (with up to ~380,000 copies/µg RNA) in brain, AF (~33,000 copies/µg RNA), PF (~20,000 copies/µg RNA), and muscle (~13,000 copies/µg RNA) (Figure 3). In the same way, *ASIC* transcripts were found in high amounts in the brain (>135,000 copies/µg RNA). In the remaining tissues, high levels of *ASIC2* were mainly present in AF (~14,000 copies/µg RNA), DF (~8000 copies/µg RNA), skin (~11,000 copies/µg RNA), muscle (~7000 copies/µg RNA), and HK (~18,000 copies/µg RNA) (Figure 3), while *ASIC*4 levels were high in fins, muscle, and HK (Figure 4b and Figure A1b).

*PIEZO2* was analyzed with three transcript variant-specific primer pairs. The primer pairs 1 and 4 are located on an alternatively spliced *PIEZO2* variant, which is specific to neurons in mammals [63]. Primer pair 3 is specific for two *PIEZO2* transcript variants including one that has not been exclusively described for neurons in mammals [63]. Transcript variant 1 was strongly expressed in AF and brain (~9200 to ~12,000 copies/µg RNA) (Figure 3). Transcript variant 4 was detectable in the AF at a level of~2000 copies/µg RNA, and to a lesser extent in the other examined tissues (>1000 copies/µg RNA in PF, skin, and muscle; <100 copies/µg RNA in DF, liver, HK, and brain) (Figure A1b). The transcript variant 3 was strongly expressed (>35,000 copies/µg RNA) in the PF, DF, brain, and, to a significantly lesser extent (~6000 copies/µg RNA), in the AF (Figure A1b).

### 3.3. Comparison of Gene Expression between Salmonid Species

In addition to the expression analysis of neuron- and glial-cell marker genes in *O. mykiss*, the expression of a subset of these genes was profiled in *C. maraena*. Here, a generally lower copy number level than in *O. mykiss* was found (Figure A2). *TRPC1* showed a congruent expression pattern in *C. maraena* and *O. mykiss*, but in the latter species, it was higher expressed by a factor of 10. The general neuron-marker genes *NEFL* and *NGFR* showed almost similar expression patterns between the tissues of both species, but the expression levels are higher in the adipose fin of *O. mykiss* by the factor 42.6 and 6.3, respectively, in comparison to *C. maraena*. Interestingly, the *PIEZO2* primer pair 1 generated considerably higher levels in *C. maraena*. Moreover, this transcript variant was more strongly expressed in *C. maraena* in the dorsal fin than in the adipose fin, whereas the opposite was observed in *O. mykiss*.

## 4. Discussion

### 4.1. Gene-Expression Profiling Indicates the Innervation of the Salmonid Adipose Fin

We established qPCR assays for 20 genes specific for neurons and glial cells (cf. Figure 1 and Figure 4a). The obtained qPCR data suggest the presence of nerve fibers in the adipose fin. This is indicated by specific gene-expression patterns of glial cells that are generally absent in tissues without direct association to neurons [51]. The transcripts coding for ten mechanoreceptors (cf. Figure 1 and Figure 4b) have been selected to cover a wide range of receptors known from vertebrates. The AF showed the most prominent expression of mechanoreceptors compared with the innervated tissues brain, skin, and muscle, all of which have well-defined mechanoreceptive potentials. 

*NEFL* and *PRPH* are the most widely used marker genes for small axons [45,64]. The reception spectrum of fibers expressing these markers covers sensations from nociception to mechanoreception [35,65]. *NEFL* and *PRPH* transcripts were highly abundant in all three fin types analyzed. On the other hand, they were expressed only at low levels in brain, skin, and muscle, suggesting that other kinds of neurons are present there. *PVALB*7 is a marker for very large, strongly myelinated neurons [66,67] and was outstandingly high-expressed in brain and skin, but showed low expression values in fins. This might indicate that large nerve trunks do not innervate the fins. 

*GFAP* is a marker for astrocytes in the CNS and Schwann cells in the PNS [68,69,70,71]. *GFAP* was most abundantly expressed in the brain and in AF. This is in line with findings from Buckland-Nicks and colleagues [23], who identified plenty of GFAP-positive cells within the AF using antibody staining. The association of GFAP-positive cells, nerve cells, and collagen was described by Buckland-Nicks [23] as common for receptor structures.

*NGFR* and *SOX*10 are highly specific marker genes of innervated tissues [37,51,65,72,73]. Both were unanimously and significantly lower transcribed in the liver and HK compared with brain and fins. The *NGFR* gene was even more highly expressed in the AF than in the DF or even the brain, indicating the presence of nerve structures [51,65,72,73,74].

PIEZO2 is known as key mechanotransducer, particularly in sensory afferents [39]. Orthologs in mammals and fish show a high degree of conservation of the nucleotide (nt) sequences and exon.

Borders (Figure A3). Accordingly, we found abundant PIEZO2 copy numbers in all fins, with varying copy numbers between the different salmonid-specific transcript variants. ASIC2, TPRC1, and KCNK2 build up channel proteins with mechanoreceptive function. ASIC2 mainly occurs in mechanoreceptive afferents. TPRC1 is responsible for mechanoreception in a tactile and contact-related manner [46]. KCNK2 is described as being physiologically important for tuning the activation of mechanoreceptive DRG neurons [75]. These three genes were expressed in the adipose fin to a much greater extent compared to all other innervated tissues except the brain.

### 4.2. The Expression of Neuron- and Glial-Cell Markers Is Tissue-Specific in Salmonids

We recorded tissue-specific expression patterns for most of the investigated genes, which are putatively involved in the proprioceptive machinery in the muscle. 

The muscle tissue of rainbow trout expressed relatively high levels of *ASIC1*, *TRPC1*, *KCNK10,* and *CACNA1H*. Additionally, the copy numbers of all *ASIC* and *PIEZO2* variants were detected at substantially high levels. Moreover, the copy numbers of *SLC1A2*, a glutamate transporter, and *TPH2*, the rate-limiting enzyme in the serotonin synthesis [76], were at high concentrations. Glutamate has several well-known and proposed functions in the muscle tissue. On the one hand, it acts as neurotransmitter within the muscle spindles [77]. On the other hand, it might be metabolized in the muscle, and SLC1A2 is necessary for its transport [78]. TPH2 is vital for efferent γ-motor neurons using serotonin in the sensory feedback of muscle spindles [79]. ASICs are involved in mammalian muscle spindle mechanotransduction [80,81], and PIEZO2 is considered as the principle mechanotransducer in proprioception [67]. Taken together, these genes indicate the presence of mechanoreceptive muscle spindles. Confirmatory, markers for nerves and glial cells, in particular, *PRPH*, *NGFR*, *GFAP*, *MBP*, *MPZ,* and *PMP22*, were also expressed in substantial levels in the muscle of rainbow trout. Furthermore, the strong expression of *NCAM1* and *SOX10* in the muscle indicates a higher density of glial cells, which are necessary for large nerve fibers. *PVALB7*, which is required in innervating muscle spindles with large neurons [67], was present in the muscle in similar high copy numbers as in brain. 

The skin tissue of the rainbow trout shows a different expression pattern compared to all other tissues and appears to be interspersed with large nerve strands. This is consistent with the knowledge about the nerve supply of the skin of higher vertebrates [33,34,35,44]. The cutaneous low-threshold mechanoreceptors (LTMRs), responsible for touch sensitivity in vertebrates, possess large and highly myelinated neurons that require correspondingly high amounts of glial cells. In the skin of mammals, particularly high proportions of glial-cell-specific genes *MPZ*, *MBP*, *PMP22,* and *S100B,* as well as the neuron marker *PVALB7,* are expressed. This agrees with the results of this study on rainbow trout. However, only few copies were detected for the mechanoreceptive channel proteins (necessary for LTMRs), except for TPRC1, KCNK2, ASIC1, and ASIC2, and the modulator CACNA1H. We note that mechanoreception in the skin requires several other receptor proteins that have not been included in the present study.

In HK tissue, many specific nerve markers, such as *TUBB3*, *STMN2*, *MAP2*, and *SULT4A1*, revealed particularly high expression levels. The teleost HK is a lympho-myeloid compartment containing immune and endocrine cells, which secrete cortisol, thyroid hormones, and catecholamines [82], such as dopamine. Serotonin is known to stimulate the secretion of cortisol in fish [83]. In this context, we refer to two important non-immune cells with different origin, the chromaffin cells, and the interrenal cells [84]. The chromaffin cells are descendants of neural crest cells, which share many functions and secretion patterns with peripheral neurons and glial cells [85]. Above all, *ASICs* and *PIEZO1* were strongly expressed in the HK. Both gene products are known to be involved in the fluid balance of teleost cells [86]. Of note, PIEZO1 is not associated with nerve cells and was considered rather as a reference gene in this study. In addition, the genes *TH*, *TPH2,* and *SCL17A7*, coding for enzymes involved in the serotonin and dopamine synthesis and the transport of glutamate, respectively, were strongly expressed in HK. The cell adhesion molecule NCAM1 is responsible for maintaining glial-neuronal connections [87] and has vital functions in natural killer cells and dendritic cells [88]. Both immune-cell populations are abundantly present in the HK since this organ is the main hematopoietic organ in fish [84]. *TUBB3*, a microtubule-forming gene, was included in this study as another reference gene, since its transcripts are not transported to the axons [41]. This supports our observation that the HK has by far the highest concentration of *TUBB3* transcripts compared with the more innervated tissues. 

In the liver, there is virtually no expression of any receptor channel protein. Only *PIEZO1* was detectable at higher levels. Besides, *SLC17A8* encoding a glutamate transporter was highly expressed. Glutamate transporters allow the uptake of glutamine and glutamate into the liver cells, where glutamate is involved in amino-acid metabolizing pathways [89]. 

The genes *NTRK2*, *NTRK3,* and *GFRa2* encode neurotrophic receptors and were used in this study to distinguish between the different nerve types, as previously done in studies on mammalian models [32,33,43,45,63,67]. Neurotrophins control the differentiation and survival of nerve cells, whereby different classes of neurons depend on different neurotrophins [90]. However, the present study revealed remarkably high levels of neurotrophin-encoding transcripts in the liver and, therefore, neurotrophins might have a cross-tissue function. 

### 4.3. Mechanosensation Is a Characteristic of the Salmonid Adipose Fin

The overall expression profile of the adipose fin (Figure 5) highly suggests the presence of nerve endings including mechanoreceptive channel proteins. *PRPH*, *NEFL,* and *NGFR* indicate the presence of small neurons with unmyelinated or slightly myelinated axons. This assumption is supported by the presence of the myelin-forming genes *MBP*, *MPZ* and *PMP22* in the adipose fin (compared to skin and muscle tissue, for instance), although at low levels. The suggested afferent nerve endings—defined as free nerve endings, C-fibres, C-LTMRs, and Aδ-fibers—may be coupled to collagen fibers via GFAP-positive glial cells [23,91]. These are able to sense mechanical stimuli through movements of the fin structure. *TRPC1*, *PIEZO2,* and *KCNKs* were only recently described as markers for C-LTMRs [65]. The expression profiles of the fins of rainbow trout indicate the presence of smaller mechanoreceptive C-fibres. Besides the mechanoreceptive function, it seems moreover likely that pain signals can be perceived in the adipose fin since many smaller nerve cells are known to be nociceptors. These were, however, not included in our gene panel.

## 5. Conclusions

The present study suggests that the adipose fin is innervated by a high amount of small nerve fibers with, most probably, mechanoreceptive potential. In the adipose fin of rainbow trout and maraena whitefish, the entire repertoire of neurons, glial cells, and receptor proteins seems to be present on the RNA level. This supports a previous hypothesis about the adipose fin as a flow sensor [22,23], and thus its significance for the animal’s locomotion in water currents. With regard to the welfare of fish, our data accelerate the discussion about the use of adipose-fin clipping for marking purposes. On the one hand, the adipose fin is a criterion for the choice of suitable sexual partners [94] and, on the other hand, contributes to the swimming efficiency [26]. Thus, the resection of the adipose fin tissue seems to be a less suitable method, particularly from an economic point of view regarding sea ranching and large-scale aquaculture in the future. 

## Figures and Tables

**Figure 1 genes-11-00021-f001:**
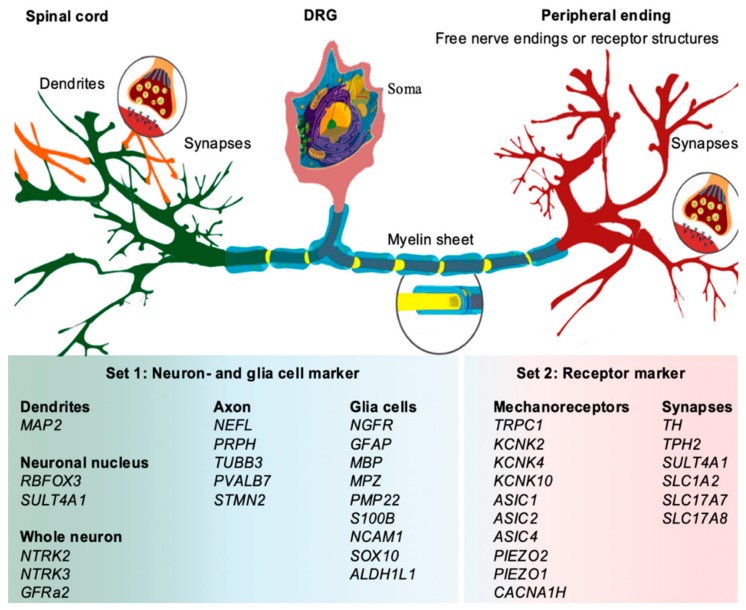
The pseudo-unipolar dorsal root ganglion (DRG) of a vertebrate neuron cell. Dendrites and synaptic connections within the spinal cord (**green**) are shown on the left side. The soma (**pink**) lies within the DRG beside the spinal colon. The axon (**yellow**) connects dendrites and soma to the peripheral ending (**red**) where the axon ramifies into free nerve endings or builds up receptor structures like Merkel, Ruffini, Meissner, or Pacinian corpuscles. The signal transmission between the peripheral ending and receptor structures is similar to synapses, which use equal neurotransmitters. The axon can be enveloped by myelinating Schwann cells (**turquoise**). The panel of genes selected for the present study and the expected site of translation are listed below the illustration. The graph is based on a free illustration (Wikipedia) by Mariana Ruiz Villarreal.

**Figure 2 genes-11-00021-f002:**
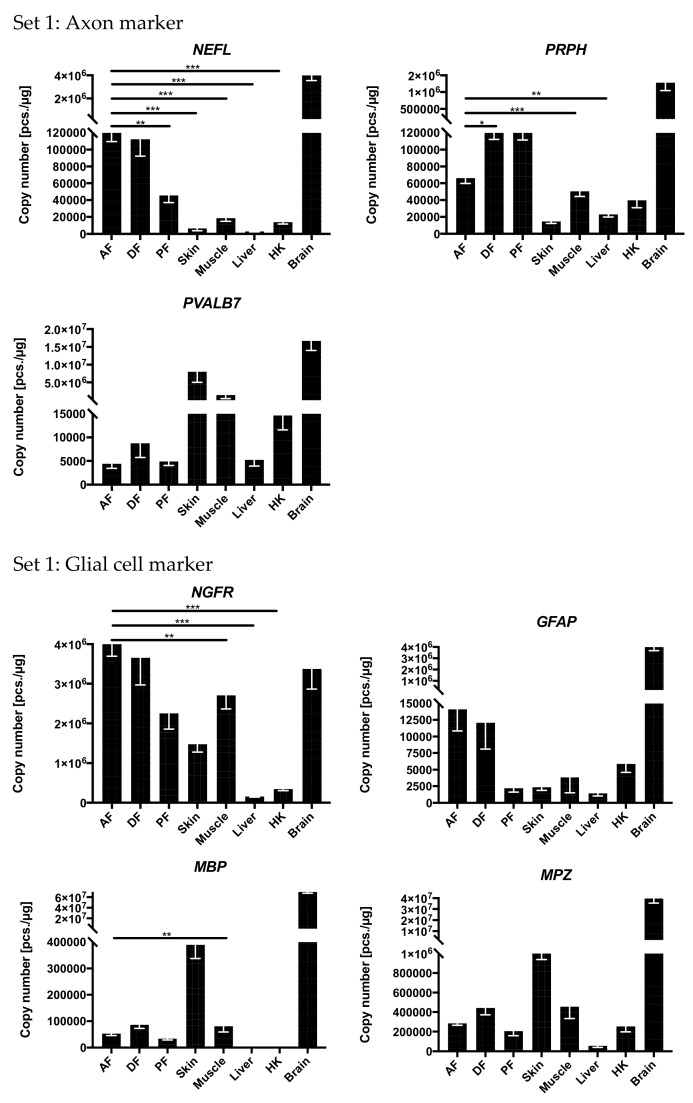
Expression levels of selected neuron and glial cell marker genes from Set 1 across tissues. The expression levels are given as absolute copy numbers (per 1 µg RNA) normalized against three reference genes. Statistically significant deviations are indicated only between AF and the other tissues. Expression values determined in brain were excluded from the statistical evaluation. Significance levels are indicated by * *p* ≤ 0.05, ** *p* ≤ 0.01, *** *p* ≤ 0.001. Error bars indicate the SEM.

**Figure 3 genes-11-00021-f003:**
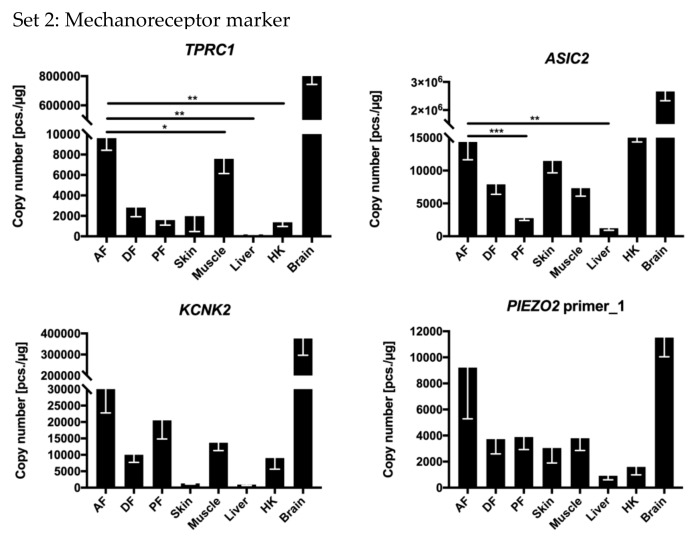
Expression levels of selected mechanoreceptor marker genes from Set 2 across tissues. The expression levels are given as absolute copy numbers (per 1 µg RNA) normalized against three reference genes. Statistically significant deviations are indicated only between AF and the other tissues. Expression values determined in brain were excluded from the statistical evaluation. Significance levels are indicated by * *p* ≤ 0.05, ** *p* ≤ 0.01, *** *p* ≤ 0.001. Error bars indicate the SEM. The presentation is limited to those genes which have proven to be particularly significant and meaningful in literature research, all others are listed in Appendix A
Figure A1b and Figure 4b.

**Figure 4 genes-11-00021-f004:**
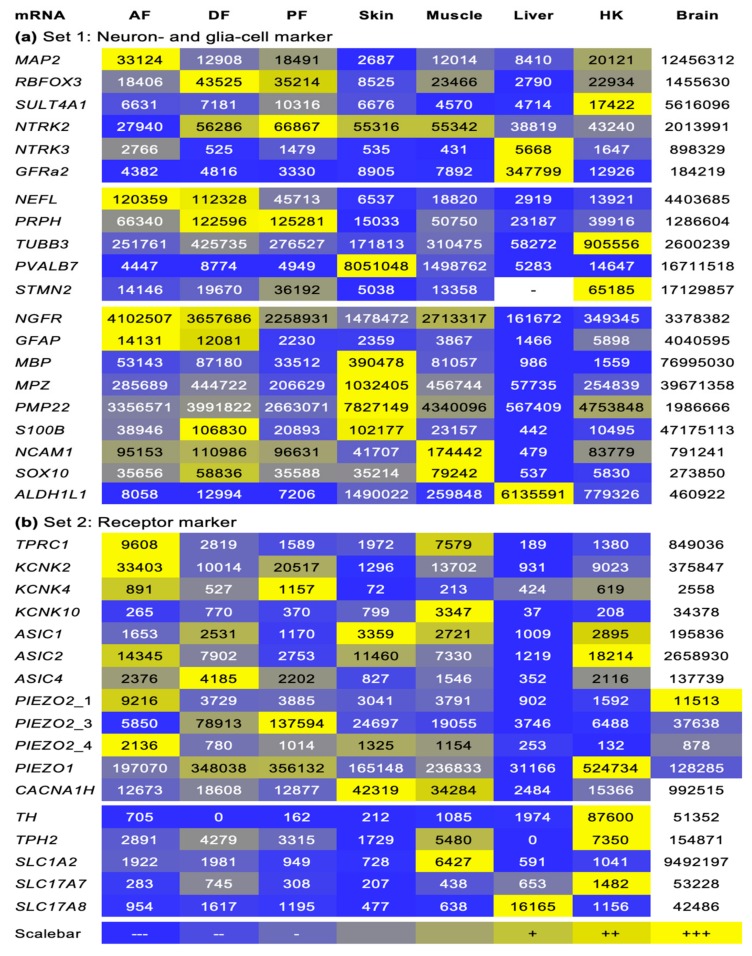
Expression profile of (**a**) neuron and glial cell- and (**b**) mechanoreceptor-specific marker genes across all the tissues investigated in *O. mykiss*. Field numbers indicate the absolute copy number per 1 μg RNA. Color codes range from low abundance (dark blue) to high abundance (bright yellow) relative to the mean expression value of each particular gene. Expression in the brain was mostly excluded from the HeatMap illustration due to extremely high expression levels.

**Figure 5 genes-11-00021-f005:**
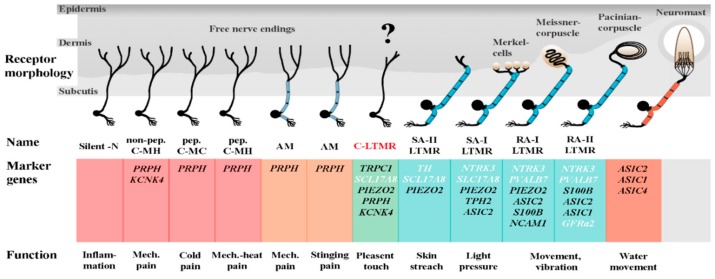
Summary of the possible presence of somatosensory receptors in the adipose fin of rainbow trout. Listed are the genes indicating the presence of nociceptors with free nerve endings (red, orange) and with specialized receptor structures (green) and cutaneous LTRMs with receptor corpuscles (blue) as well as neuromasts (light red). Marker genes identified in the adipose fin in ample amounts are printed in black. Marker gene names are printed in white if the expression in the AF was not outstandingly high in the comparison of the tissues analyzed. The specific cell types for which the listed genes are characteristic are labeled as follows: Non-peptideric C-fibre mechano-heat receptor (non-pep.-C-MH), peptideric mechano-cold nociceptor (pep. C-MC), peptideric C-fibre mechano-heat nociceptor (pep. C-MH), A-fibre mechanonociceptor (AM), C-fiber low-threshold mechanoreceptor (C-LTMR), Aβ-fiber slowly-adapting type I and type II LTMR (SA-I LTMR and SA-II LTMR), Aβ-fibre rapidly-adapting type I and type II LTMR (RA-I LTMR and RA-II LTMR). Marker genes were extracted from literature [35,36,37,38,39,40,41,42,43,44,45,46,47,65,92,93], figure is adapted from [35].

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
