# Peer review of "Gene Profiling in the Adipose Fin of Salmonid Fishes Supports Its Function as a Flow Sensor"

_genes, 2019, doi:10.3390/genes11010021_

Round 1

Reviewer 1 Report

This manuscript presents further evidence that the salmonid adipose fin, previously thought to be vestigial, may in fact be a mechanosensory organ with important functional roles. The evidence is based on the relatively high mRNA expression of a range of mechanosensory genes compared to other tissues. The authors also point out that removal of adipose fins, standard practise in some restocking and aquaculture operations, is a welfare issue due the potential physiological importance of this tissue as well as the likelihood of pain perception.

In general the text is clear, although there are sentences where the English could be improved. The conclusions are supported by the results and the results are presented clearly.

There are are however two points that should be addressed and clarified by the authors.

It should be stated whether the adipose fins or any other relevant tissues were skinned before processing.

The qPCR normalisation methodology for mRA expression analysis is not clear. It is mentioned in the methods that housekeeping reference genes were analysed, but results are presented normalised to mass of input RNA, not reference genes. I can understand why this should be the case since reference genes are likely not to be equally expressed across all tissues. However the use of input RNA for normalisation does not account for any technical variation in cDNA synthesis efficiency. Use of rRNA reference genes and randomly primed cDNA synthesis might be preferable. I am however convinced by the results, technical variation appears relatively low, and the differences between tissues are clear. I recommend that the methods and results are checked for consistency, and that a clear description of the normalisation procedures are included.

Reviewer 2 Report

This is a interesting article conceptually. The authors make use of gene expression studies to explore the structure and function of adipose fins in salmonid fishes and their work sustains a critical view about the proceedings used in aquaculture. They explain that these fins are used to mark fishes from aquaculture, however these proceedings are not equivalent to other fin-clipping proceedings. In fact, by proving evidences that these fins are essential sensory structures for the fish, they make clear that the clipping is going to interfere with fish health, behavior and welfare, which has ethic but also economic implications.

The article is elegantly written and the data presented in a very clear way. I would suggest an extra read to solve some "punctuation" issues. I also suggest the use of "qPCR" always, instead of "RT-qPCR". The data from qPCR is presented in number of copies/ug, however it is also nice to present it as Relative Gene Expression by calculating the 2ACt, which allows to follow in a easier way the fold-changes.
